# Research on the Influence of Exoskeletons on Human Characteristics by Modeling and Simulation Using the AnyBody Modeling System

**Lin Cao, Junxia Zhang \*, Peng Zhang \* and Delei Fang**

Tianjin Key Laboratory for Integrated Design & Online Monitor Center of Light Design and Food Engineering Machinery Equipment, Tianjin University of Science &Technology, Dagunan Road, Tianjin 300222, China; caolin@tust.edu.cn (L.C.); fangdelei@tust.edu.cn (D.F.)
\* Correspondence: zjx@tust.edu.cn (J.Z.); tust2019zp@163.com (P.Z.)

**Abstract:** Lower limb-powered exoskeletons can help rehabilitate patients with lower limb disabilities. However, the changes in the biomechanical load on the human body when exoskeletons are used are still poorly understood. The goal of this study was to investigate the changes in kinematic and biomechanical parameters of the lower extremity exoskeleton when worn by normal subjects and patients with unilateral motor impairment using a virtual prototype. The effect of wearing the exoskeleton on gait was derived, and the basis for exoskeleton optimization was given. Virtual prototyping is a cost-effective method to validate the performance of exoskeleton robots. Therefore, two models, a human-exoskeleton model and an asymmetric movement disorder (SSP) subject-exoskeleton model, were developed in AnyBody software for this study. The human-exoskeleton model was driven by the kinematic data of 20 healthy participants walking in an exoskeleton at normal speed (3.6 km/h). As a comparison, the SSP subject-exoskeleton model was driven by data from five SSP subjects walking in an exoskeleton. The experimental results show that after wearing the lower limb exoskeleton, the walking gait satisfies the normal human gait characteristics, but some of the muscle forces suddenly increase. The maximum activation level does not exceed 1, which means that the exoskeleton does not cause muscle damage or fatigue in a short period of time. In both models, the vertical ground reaction force (GRF) Z correlation was the strongest ($R > 0.90$). The center of pressure (COP) X trajectory correlation was the weakest ($R < 0.35$). These findings will support the study of the effects of exoskeletal optimization. Also, some gait characteristics of exoskeletons worn by patients with unilateral dyskinesia can be initially explored.

**Keywords:** modelling and simulation; virtual prototyping; exoskeleton

## 1. Introduction

Stroke is characterized by a high incidence and disability rate [1–3], which can place a heavy burden on patients and their families. The signs and symptoms after stroke onset are usually unilateral lower limb motor deficits [4]. The current approach to stroke rehabilitation relies on the joint participation of rehabilitation physicians and physical therapists, a rather lengthy and expensive process, and the most critical problem is the huge shortage of rehabilitation professionals [5]. However, rehabilitation robots can effectively and economically provide rehabilitation to physically disabled patients compared to the traditional manual assistance of rehabilitation physicians. The rehabilitation effects of lower limb rehabilitation robots have been widely recognized [6,7]. The biomechanics of human–exoskeleton interactions influence the redistribution of load and unload in body regions and are critical to the safety and effectiveness of the device [8]. However, if exoskeletal robots are not designed properly, long-term use can cause harm to the human body [9,10]. To validate the effectiveness of exoskeletons, the traditional engineering approach is to create a simple physical model for testing and improve the design by



iteratively building a new prototype, but the cycle time of this approach is too long. There would be a significant potential risk if prototypes were tested directly on patients. In addition to long-term and short-term experimental studies, musculoskeletal models can be used for simulation-assisted exoskeleton development, optimization, and evaluation, and can be an important tool for quantifying work-related load and exoskeletal impact.

Musculoskeletal models are only able to calculate muscle forces and joint reaction forces from the input of a given motion. The field of musculoskeletal modeling has evolved a lot in the last two decades. There are many musculoskeletal modeling and simulation software available. Among the most popular ones are OpenSim and AnyBody Modeling System [11,12]. In the simulation system, the human body wears an exoskeleton to perform various movements and collects data on the interaction between the exoskeleton and the human to evaluate the performance of the exoskeleton in a more cost-effective and time-efficient manner. Ferrati used OpenSim to compare the changes in joint moments with and without the exoskeleton [13]. However, the simulations did not consider the properties of the muscles. Similarly, Zhu et al. used OpenSim and ADAMS to model the exoskeleton. The objective was to measure the alignment of the joints between the human body and the exoskeleton. A fitted kinetic equation was also developed to calculate the joint moments based on the virtual prototype [14]. Compared to OpenSim software, AnyBody has an open-source code model library. With the AnyBody model library, users can use and modify new models more easily, which greatly simplifies modeling. Stambolian used AnyBody software to simulate the process of moving an object with the human body [15]. A comparative analysis of the reaction law of the object on the human body was performed by varying the weight of the object. Studies have used simulation software to output the joint torques of exoskeletons and evaluate joint activity control performance [16–25]. Priyanshu et al. designed a virtual prototype set-up of a rehabilitation exoskeleton by merging computational musculoskeletal analysis and AnyBody simulation technology [16]. Agarwal used AnyBody simulation to study and analyze the performance of the lower limb exoskeleton. The experimental results showed that enhancement through the exoskeleton can lead to a significant reduction in muscle loading [17]. In order to make the structure of the lower limb exoskeleton more compact and simpler and provide a theoretical basis for the dynamic support effect of the lower limb exoskeleton, Li used AnyBody to simulate the knee joint lower limb exoskeleton. The simulation experiment results show that after wearing the lower limb exoskeleton, its energy consumption can be reduced by 58.6% when bearing an additional 100Kg payload. This proves that the knee-joint lower limb exoskeleton can effectively help the human body bear heavy loads and reduce the energy consumption of the human body during the walking process [21].

Some researchers have worked on the optimal design of exoskeletons using AnyBody software [26–37]. Geonea presents a design solution for a new exoskeleton robot leg mechanism. A virtual prototype based on AnyBody was created. Kinematic and kinetic analysis of the proposed exoskeleton robot was carried out. The proposed exoskeleton was found to achieve similar motion patterns in the hip and knee joints as in normal human walking [23]. Vighnesh proposed a parameter optimization method for the exoskeleton. A validation analysis was also carried out using AnyBody software. The optimized exoskeleton reduces the estimated muscle force by an average of 5.73%. The average reduction in muscle force was 14.5%, and the peak reduction was 32.2% compared to not wearing the exoskeleton [24]. Shan applied AnyBody software to establish the overall model of the human body and hip rehabilitation device. The inverse dynamics simulation and muscle force analysis of the overall model were carried out to obtain the force and muscle activity of the hip muscle groups when using the hip rehabilitation device. The experimental results were used to provide data support for the optimization of the hip rehabilitation device [26]. Xiang investigated the effectiveness of lower limb exoskeletons in rehabilitation training [28]. The contraction rates of the major muscles of the lower limbs were analyzed by simulating the trajectory of the rehabilitation robot in the sagittal plane. The results showed that the contraction rates of the thigh muscle groups were the same. The differences in the calf and

hip muscle groups were statistically significant. However, the analysis targeted a single factor and was not comprehensive enough. Geonea proposed a method for modeling an exoskeleton robot based on AnyBody software. The dynamics of the robot system are analyzed using AnyBody software and ADAMS. The relationship between the change law of the turning angle of the kinematic joints, the connection force of the kinematic joints, and the GRF is given. Finally, a solution for easy manufacturing of the robotic system using 3D printing technology was proposed [29].

Although many studies have explored human body movement patterns and optimization strategies of exoskeletons [38–44], there is a lack of human–machine simulation studies of a real exoskeleton attached to the musculoskeletal model with unilateral movement disorders. At present, most studies have been performed on normal people. Most stroke patients exhibit unilateral dyskinesia [45–47]. Since exoskeletons are intended for patients with disabilities, it is necessary to explore whether exoskeleton robots could actually meet the needs of patients. In this paper, two virtual models were built in AnyBody software. The first one is a human-exoskeleton model driven by healthy slow walking kinematics. The kinetic and biomechanical parameters of the human model and the human exoskeleton model were compared and analyzed. The effect of the lower limb exoskeleton on the human body can be quantitatively analyzed. The second model is the SSP subject-exoskeleton model, which is driven by kinematic data from subjects with unilateral movement disorders walking with a real exoskeleton. Since the measurement data were obtained from participants wearing the exoskeleton, the predicted GRF accuracy will validate both models. The predicted GRF and COP trajectories of the human exoskeleton model and the SSP subject-exoskeleton model were also analyzed, which is important to investigate the differences between normal human and SSP subjects when wearing an exoskeleton at a normal speed of 3.6 km/h. The results of the study will support the optimal design of the exoskeleton robot to make it more suitable for SSP patients.

Based on the background and research questions, the research hypotheses of this study mainly include the following:

1. The gait parameters and biomechanical load of the human body change after wearing a lower extremity exoskeleton.
2. There are differences in the changes in gait parameters and biomechanical parameters between the exoskeleton worn by patients with unilateral dyskinesia and those worn by normal subjects.
3. By studying the changes in gait parameters and biomechanical parameters of the exoskeleton, the design of the exoskeleton can be optimized.

## 2. Model Development

### 2.1. The Proposed Novel Lower Limb Exoskeleton Robot

The biomechanical characteristics of humans provide a reference for the structural design of the exoskeleton. The developed lower limb exoskeleton is shown in Figure 1a,b. The introduction of each component of the lower limb exoskeleton robot is shown in Table 1. The exoskeleton designed in this paper mainly considers motion in the sagittal plane. For each leg, it has one active DoF for the hip joint and knee joint, respectively, and one passive DoF for the ankle joint. The limit of flexion in the hip joint is $0°$, and the maximum extension angle is $70.18°$. The limit of flexion in the knee joint is $102.53°$, and the maximum extension angle is $0°$. The hip joint and knee joint were driven by a direct current servo motor, and the parameter values of joint power elements are shown in Table 2. The ankle joint adopted a passive reset function. The step function is utilized to control the speed of the ball screw. The linear motion of the ball screw is transformed into the rotational motion of the hip joint and knee joint in the sagittal plane, which approximately fits the normal gait of a human. This exoskeleton fits patients who are 1.50 m to 1.90 m tall, which covers more than 99% of corresponding adults, with a maximum bodyweight of 100 kg.

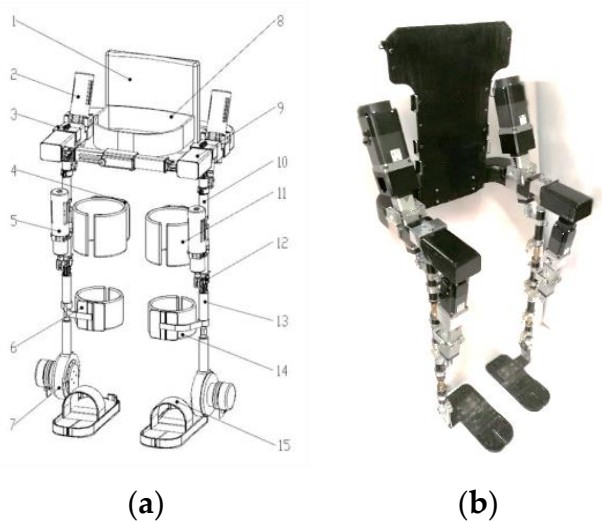

**(a)**                    **(b)**

**Figure 1.** Lower limb exoskeleton. (**a**) (1) Back support; (2) Driving source of hip joint; (3) Hip joint component; (4) Brace of thigh; (5) Driving source of knee joint; (6) Brace of calf; (7) Ankle joint component; (8) Flexible belt of waist; (9) Waist component; (10) Thigh component; (11) Flexible belt of thigh; (12) Knee joint component; (13) Calf component; (14) Flexible belt of calf; (15) Pedal. (**b**) Lower limb exoskeleton prototype.

**Table 1.** Introduction of each component of the lower limb exoskeleton robot.

| Component | Material | Mass/Kg |
|---|---|---|
| Back support | Carbon fiber | 1.2 |
| Hip joint component | High strength aluminum alloy | 1.5 |
| Knee joint component | High strength aluminum alloy | 1.2 |
| Ankle joint component | High strength aluminum alloy | 1.2 |
| Brace of thigh | Carbon fiber | 2.4 |
| Brace of calf | Carbon fiber | 2.4 |

**Table 2.** Parameter values of joint power elements.

| Power Components | Specification | Hip Joint | Knee Joint |
|---|---|---|---|
| Motor | Rated output (W) | 750 | 400 |
| | Rated torque (N-m) | 2.4 | 1.3 |
| Ball Screws | Stroke range (mm) | 100 | 100 |
| | Maximum dynamic load (N) | 4000 | 2000 |

*2.2. Musculoskeletal Model Construction*

The lower limb exoskeleton model was imported into the AnyBody software, just as shown in Figure 2a. It was controlled by the program 'FitnessMachine.any'. The human musculoskeletal model and the lower limb exoskeleton model were connected by the program 'JointsAndDrivers.any'. The human-exoskeleton model is shown in Figure 2b. The human musculoskeletal model was built based on the application repository model library. The motion data captured by the Vicon system. The data were imported into AnyBody software through the "C3D to Anyscript conversion program." The inverse kinetic method was used to calculate muscle force, joint force, and joint moments. The motion of the exoskeleton was constrained by human kinematic data. The ankle and hip joints of the exoskeleton are aligned with the human body, respectively. The knee joint of the exoskeleton is aligned with the human body along the anterior–posterior axis. Compared to the human exoskeleton model, the freedom of flexion and extension of the left knee joint of the SSP subject-exoskeleton model was removed.

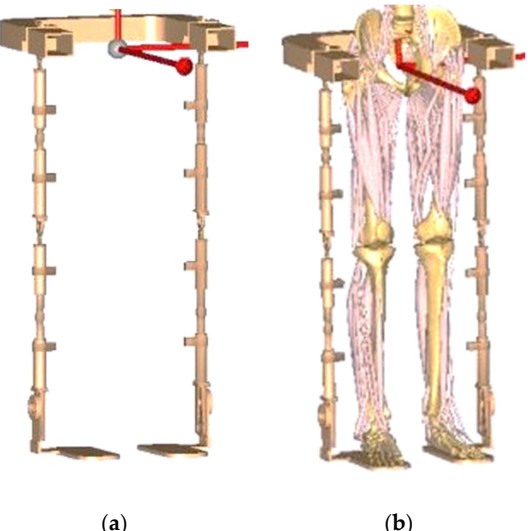

(**a**)                    (**b**)

**Figure 2.** (**a**) Lower limb exoskeleton model; (**b**) Human-exoskeleton model.

## 3. Experiment

### 3.1. Participants

Twenty healthy volunteers (males aged $24.70 \pm 1.42$ years, weight: $67.50 \pm 8.59$ kg, lower limb height: $900.30 \pm 34.96$ mm, knee joint width: $112.49 \pm 5.72$ mm, and ankle joint width: $71.99 \pm 4.15$ mm) were recruited from Tianjin University of Science & Technology. Relatively young, healthy subjects were selected for the gait analysis study, which can exclude the influence of the age factor to the maximum extent and obtain more accurate and reliable study results, providing a better reference for the next study. In order to explore the walking characteristics of SSP subjects clearly, the movement angle of the knee joint was set at 0 degrees for SSP subjects. The test was performed using an adjustable medical knee immobilization support (Medway, Shanghai, China; product size: 500 mm $\times$ 200 mm $\times$ 100 mm) and a thin wood strip with rounded corners (20 mm $\times$ 2.5 mm $\times$ 8 mm) to assist in tying the dorsal aspect of the knee joint, just as shown in Figure 3. The detailed component composition is shown in Figure 4. The purpose was to limit the mobility of the knee joint in the flexion and extension directions to simulate the joint mobility characteristics of the knee joint in a rigid gait approximation. Written informed consent was obtained from the participants, and this study was approved by the Tianjin University of Science and Technology Research Ethics Board. The study plan and experimental design meet ethical requirements, and the study is conducted with the informed consent of the participants.

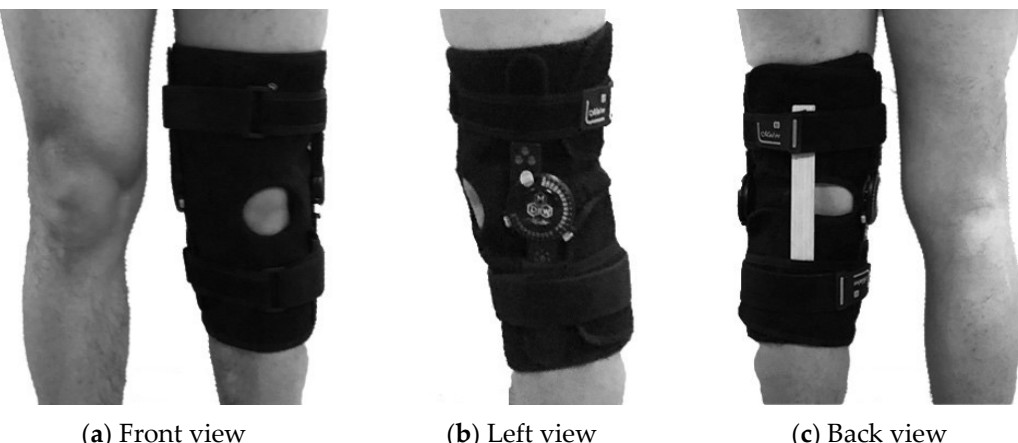

(**a**) Front view                    (**b**) Left view                    (**c**) Back view

**Figure 3.** Binding mode of the knee fixation brace.

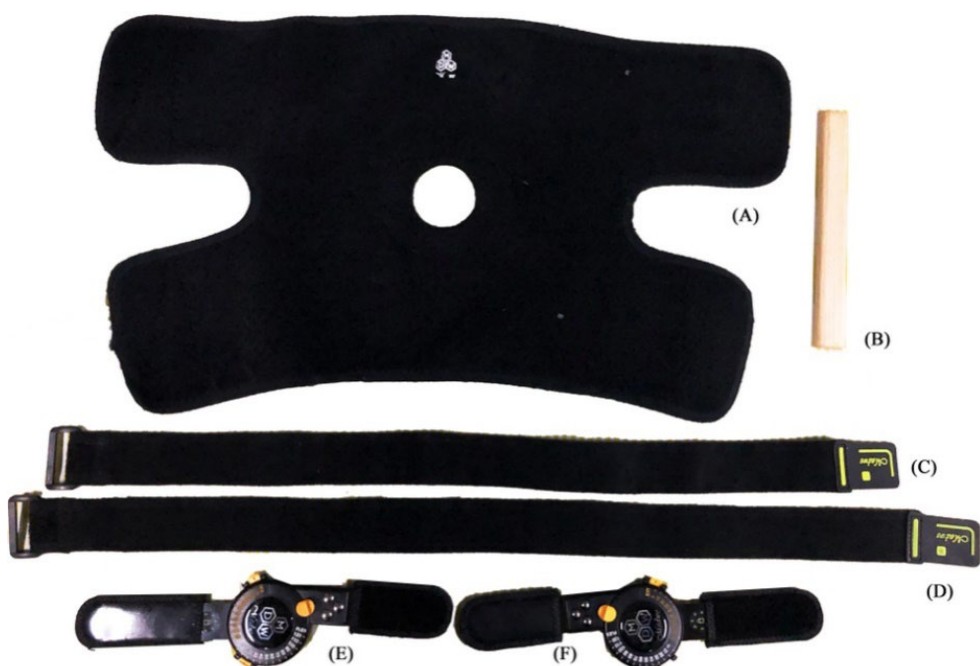

**Figure 4.** Knee fixation brace: (**A**) Protective Velcro; (**B**) Rounded corner support small wooden strips; (**C**) Lower leg Velcro straps; (**D**) Thigh Velcro straps; (**E**) Left side aluminum support chuck; (**F**) Right side aluminum support chuck.

### 3.2. Gait Experiments

The data collection platform is shown in Figure 5. The data collection platform includes Pedar-X (Novel Pedar, German), Telemyo 2400 DTS (Noraxon, America), AMTI three-dimensional force plates (AMTI-BP400600, America), and Vicon T40–S (Vicon, England). The frequency of data acquisition was set at 1.5 kHz. The adjustable medical knee fixation support has an adjustment range of 0 to 180°. The specific steps of bondage are as follows: (1) first, according to the condition of the patient's knee joint, select the appropriate product type to fit, and tie the magic pad; (2) the subject is then allowed to stand upright, with the axes of the angular chucks on each side corresponding to the lateral epicondyle of the knee on the wearing side; (3) finally, fix the size of the leg straps and add rounded thin wooden strips to adjust the looseness. The subjects were allowed to walk for 10 min after wearing the device, during which time the knee joint could not be flexed for adjustment and optimization. Due to the limitation of the knee joint of the left lower limb, the main objective of this study was to observe the kinematic and biomechanical changes in the right lower limb. Hence, the relevant parameters of the right lower limb in the sagittal plane were analyzed. The specific experimental scheme is as follows:

1. All participants were required to walk on a straight test bench (AMTI BP-400600), just as shown in Figure 5.
2. To eliminate the error between different tests, the gait cycle was standardized. A complete gait cycle is 100%. The beginning of the gait cycle, 0%, means the right heel first touches the ground. The end point 100% represents the right heel touching the ground again.
3. The walking test was repeated 5 times for each participant. Appropriate rest between sets was also given to exclude the effects of muscle fatigue.

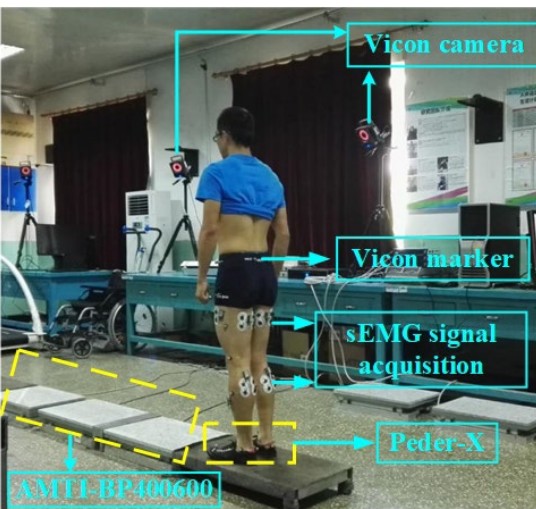

**Figure 5.** Data collection platform.

*3.3. Muscle Parameters Selection*

The selected muscles were divided into three groups include hip joint group (Biceps Femoris, Semitendinosus, Rectus Femoris), knee joint group (Vastus Medialis, Vastus Lateralis) and ankle joint group (Tibialis Anterior, Medial Gastrocnemius, Lateral Gastrocnemius). By comparing the various factors of the muscle when walking with or without an exoskeleton, the degree of muscle involvement in training can be judged. Baratta proposed a muscle mechanics model that reflected the mechanical properties of muscles through muscle tension, muscle length, and contraction velocity [39].

$$F = KS^2 + Ce^{-|v|} + F_0 \tag{1}$$

where $F$ is the muscle force, $K$ is the coefficient of the muscle tension–muscle length, $S$ is the change in the muscle length, $C$ is the coefficient of the muscle tension contraction velocity, $v$ is the contraction velocity, and $F_0$ is the initial muscle force.

*3.4. Data Analysis*

Following the previous research [48], the impact of the designed exoskeleton robot on the human body was evaluated through dynamics and biomechanics parameters. The Pearson correlation coefficient ($R < 0.35$, weak; $0.35 < R < 0.67$, moderate; $0.67 < R < 0.9$, high; $R > 0.9$, high) and root mean square error (*RMSE*) of the GRF and COP trajectory were used to evaluate the human-exoskeleton model and the SSP subjects-exoskeleton model [49–51]. The GRF peak and COP values of the two models were compared with those of the human-only model. The GRF peak includes GRF X (mid-lateral), GRF Y (anterior-posterior), and GRF Z (vertical). The COP values were calculated and averaged at the steady-state phase. To reduce the effect on the evaluation metrics $R$ and *RMSE*, the last 5% of the attitude phase was removed from the COP ballistic analysis.

**4. Results**

*4.1. Comparison between the Human-Exoskeleton Model and the Human-Only Model*

4.1.1. Joint Force Analysis

As is shown in Figure 6a–c, compared with the human-only model, for the human-exoskeleton model, the maximum bending force of the hip joint was reduced by 2.31 N/kg, and the maximum extension force was reduced by 1.68 N/kg. The maximum extension force of the knee joint was reduced by 24.15 N/kg. The peak value of the dorsiflexion muscle of the ankle joint was reduced by 28.79 N/kg. The experiment results show that the lower limb exoskeleton has the most notable auxiliary function at the knee joint.

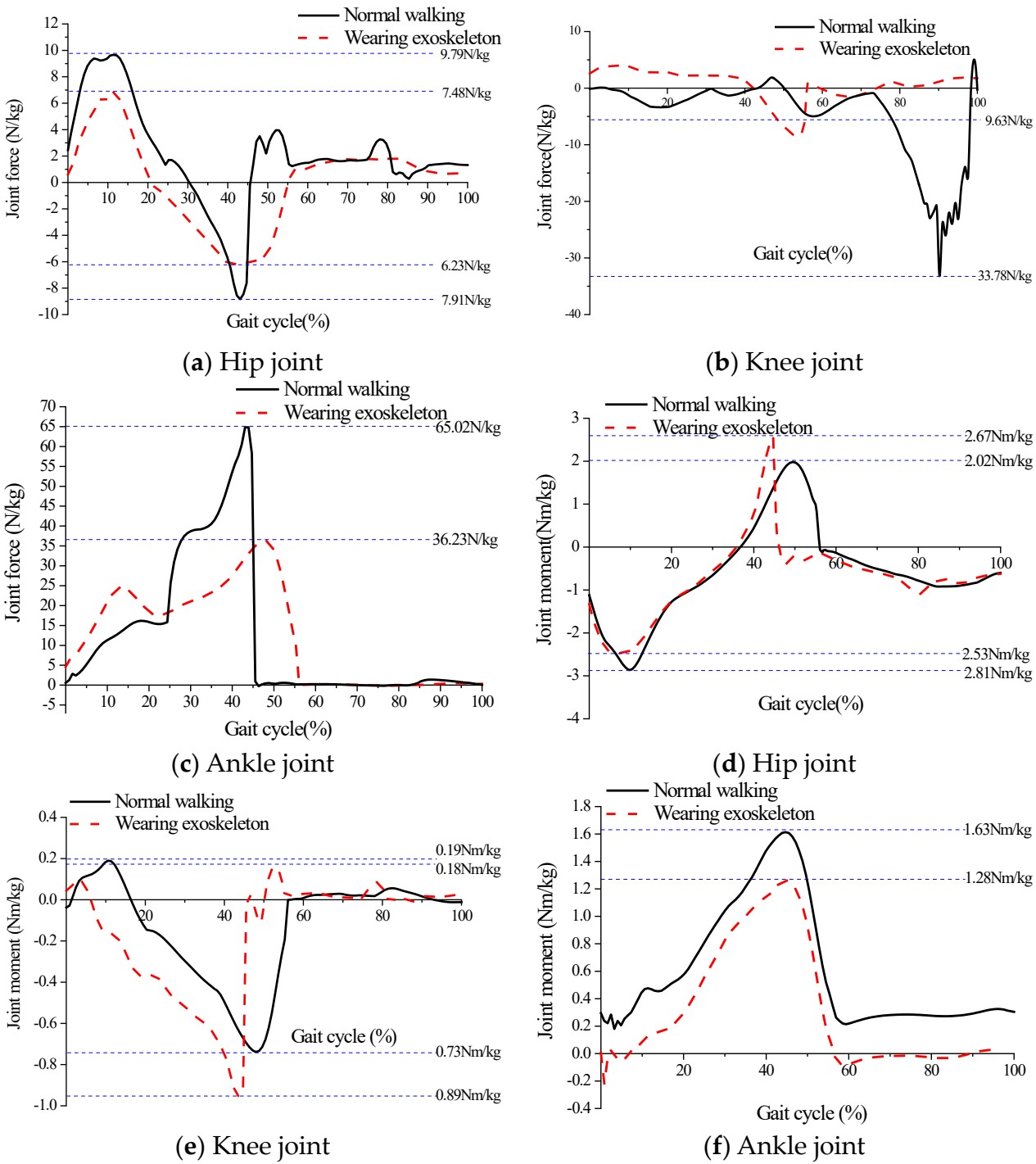

**Figure 6.** The dynamic analysis.

4.1.2. Joint Moment Analysis

As is shown in Figure 6d–f, compared with the human-only model, for the human-exoskeleton model, the maximum flexion moment of the hip joint was reduced by 0.65 N·m/kg, and the maximum extension moment was reduced by 0.28 N·m/kg compared with the human-only model. The maximum flexion moment of the knee joint was reduced by 0.16 N·m/kg, and the maximum extension torque was reduced by 0.01 N·m/kg. The maximum dorsiflexion moment of the ankle joint was reduced by 0.35 N·m/kg.

### 4.1.3. Biomechanical Parameters Analysis

The degree of muscle activation is a good reflection of the muscle's contraction status. The range of muscle activation is 0~1. When the muscle activation was greater than 1, the muscle was in a state of fatigue or muscle injury [52]. As is shown in Figure 7a–d, the maximum degree of muscle activation of each muscle does not exceed 1, i.e., the exoskeleton has little damage to the human body. The peak of the Rectus Femoris was increased by 0.456, which is the biggest increase, as shown in Figure 7a. This means that the exoskeleton is most likely to cause fatigue or injury to the Rectus Femoris.

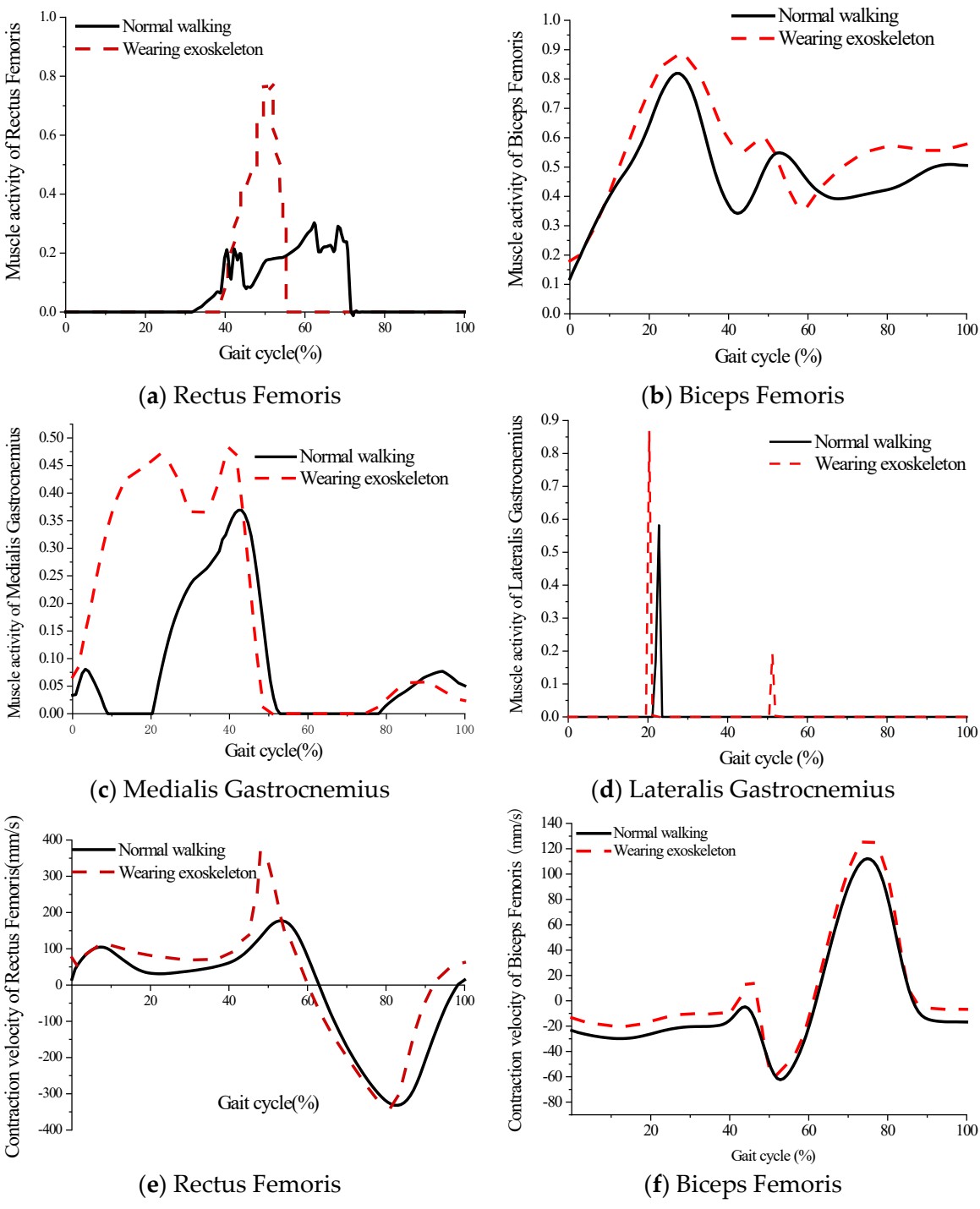

(**a**) Rectus Femoris

(**b**) Biceps Femoris

(**c**) Medialis Gastrocnemius

(**d**) Lateralis Gastrocnemius

(**e**) Rectus Femoris

(**f**) Biceps Femoris

**Figure 7.** *Cont.*

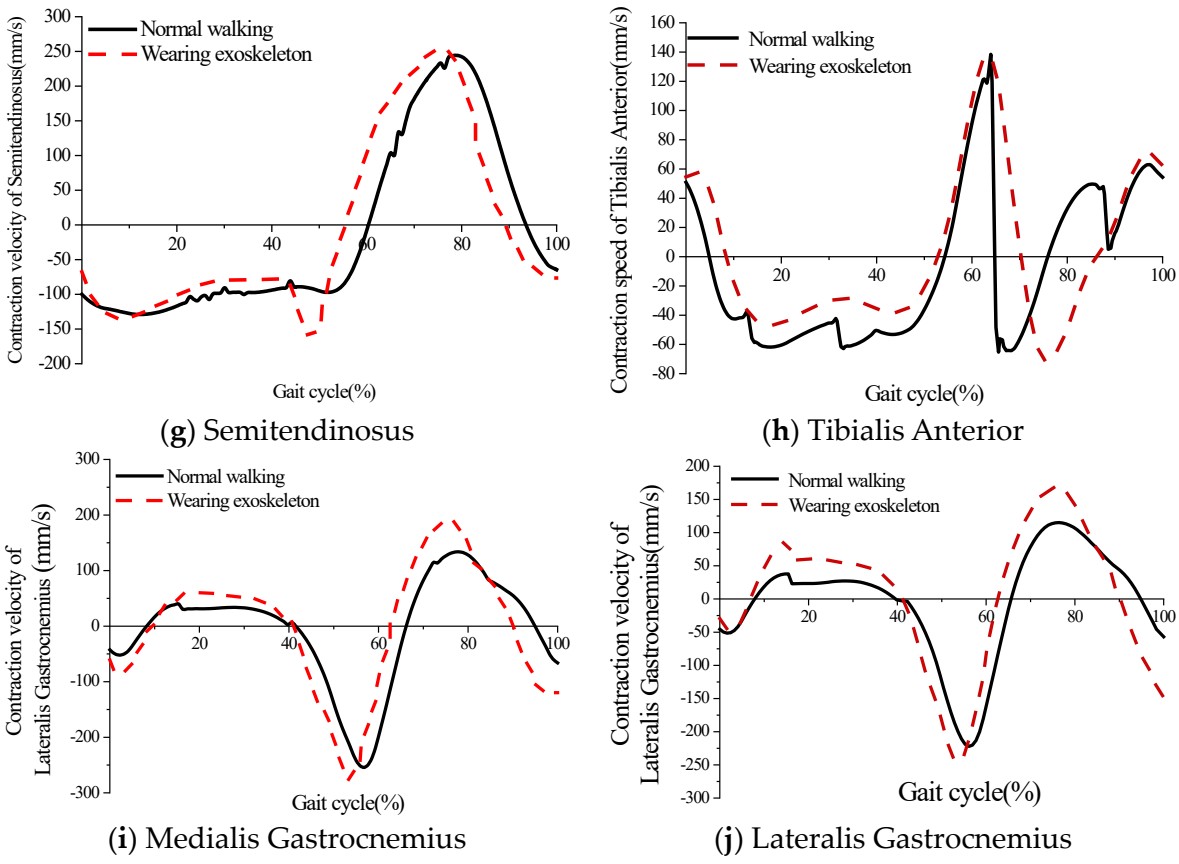

**Figure 7.** The biomechanical parameters analysis. (**a**–**d**) The muscle activity of lower limb muscles; (**e**–**j**) The contraction velocity of lower limb muscles.

The muscle contraction velocity is a good indicator of the amount of muscle force. As is shown in Figure 7e–g, compared with the human-only model, for the human-exoskeleton model, the maximum contraction velocity of the Rectus Femoris increased by 159.86 mm/s, the maximum contraction velocity of the Biceps Femoris increased by 9.18 mm/s, and that of the Semitendinosus remained unchanged. In addition, the maximum contraction velocity of the Medialis Gastrocnemius and Lateralis Gastrocnemius muscles increased by 46.68 mm/s and 38.24 mm/s, respectively, through the analysis of the results (Figure 7i,j). The muscle contraction velocity of the Tibialis Anterior muscle lagged by 8% gait cycle, but the maximum contraction velocity increased by 28.17 mm/s, as shown in Figure 7h.

### 4.2. Comparison between the Human-Exoskeleton Model and the SSP Subjects-Exoskeleton Model

The Pearson correlation coefficient is significant for all directions of the GRF and COP trajectory ($R < 0.05$). The correlation between GRF Z and COP Y is more obvious. Although the absolute value of *RMSE* is smaller on the GRF X and COP Y trajectory, the difference in the GRF peak value and COP amplitude between the human-exoskeleton model and the SSP subject-exoskeleton model is not significant enough. Except for the middle transverse GRF and the GRF Z peak at the toe, in the whole gait cycle, the GRF Z value of the SSP subject-exoskeleton model is bigger than the human-exoskeleton model. The COP X trajectory of the SSP subjects-exoskeleton model is more to the right and the COP Y is more posterior, just as shown in Figure 8d,e.

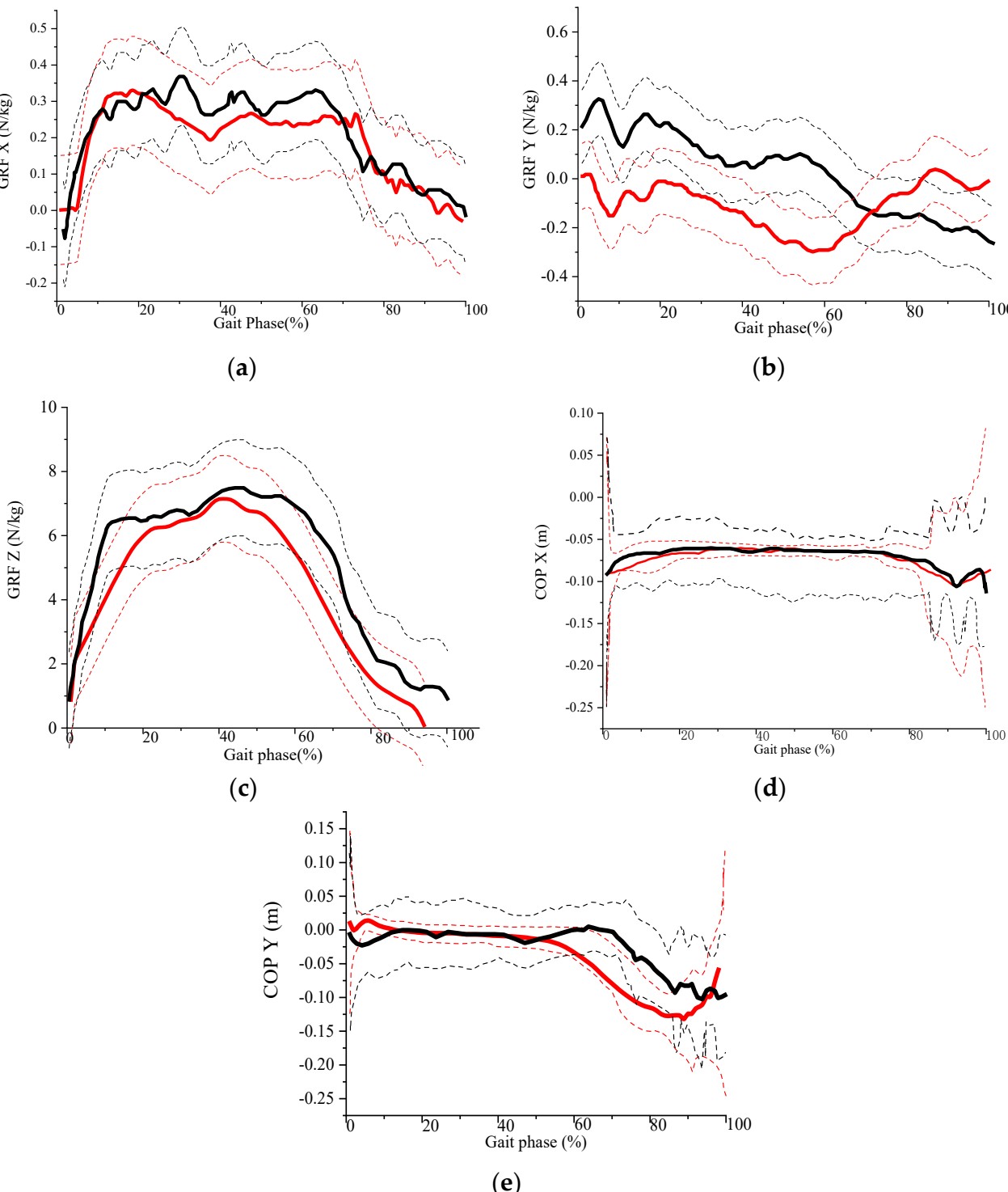

**Figure 8.** GRF and COP trajectory analysis. (**a**–**c**) SSP subjects-exoskeleton model mass normalized predicted GRF (the black thick line is the mean and the black dotted lines are the mean ± standard deviation) and human-exoskeleton model GRF (the red thick line is the mean and the red dotted lines are the mean ± standard deviation); (**d**,**e**) SSP subjects-exoskeleton model predicted COP trajectories (the black thick line is the mean and the black dotted lines are the mean ± standard deviation) and human-exoskeleton model COP trajectories (the red thick line is the mean and the red dotted lines are the mean ± standard deviation).

## 5. Discussion

The experimental results showed a bimodal pattern of muscle activity in the knee extensors. The first peak occurred at 55% of the gait cycle. At the same time, the muscle was in the centrifugal contraction phase, and the muscle contraction velocity was maximal. Therefore, there is a maximum value of muscle force, which is consistent with Hatze's velocity-tension model [53]. The second peak occurs at 70% of the gait cycle. At the same time, the muscles undergo isometric contraction. As the speed of muscle contraction increases, the muscle force gradually decreases. However, for the human exoskeleton model, the moments of the two peaks of muscle force lagged slightly behind those of the human model. The first peak is larger than that of the pure human model, while the second peak is smaller. The reason for this may be that the human body has an adaptation process when walking with the exoskeleton, which increases the interaction time between the human body and the exoskeleton, resulting in a delay in the generation of muscle force.

Peak medial gastrocnemius muscle stretch force occurs at 50% of the gait cycle when walking without an exoskeleton. Peak muscle contraction velocity occurs at 50%~60% of the gait cycle. Peak medial gastrocnemius muscle contraction force occurs at 60%–70% of the gait cycle. Peak muscle force was not consistent with peak muscle contraction velocity. The reason for this may be that the relationship between velocity and tension in previous studies was obtained with optimal muscle length (maximum isometric muscle length) [53]. However, under normal conditions, muscle length is constantly changing during walking. The kinetic and biomechanical parameters of this exoskeleton satisfy the laws of human walking and are reasonably designed. It also provides a new method for the validation of exoskeleton robots.

Despite the fact that the kinetic and biomechanical parameters of the human-exoskeleton model satisfy the normal gait, there is still an abrupt increase in muscle activation. This is due to the fixed size of the lower limb exoskeleton, which does not exactly correspond to each individual's lower limb. For this reason, a new mechanically adjustable exoskeleton mechanism was redesigned. The improved exoskeleton structure has a thigh rod length adjustment range of 470–530 mm, a calf rod length adjustment range of 350–420 mm, and a waist width adjustment range of 390–600 mm. The length adjustment principle of the thigh and calf rods is a stepless adjustment mechanism. The length of the connecting rod is adjusted by the screw driver of the double-headed stud, which is locked by the locking nut. The waist adjustment mechanism connects the two ends of the waist and the middle sleeve through a sliding guide mechanism. The left and right ends of the waist can be slid inside the middle sleeve and can be tightened and fixed by adjusting the plum nut. The length of each segment of the lower limb exoskeleton should be adjusted to suit the length of different users and to improve the coordination and unity of the exoskeleton and the wearer.

In the SSP subject-exoskeleton model, the longitudinal GRF Z correlation was the highest, while the GRF correlation was moderate in the X and Y directions. In the SSP subject-exoskeleton model, the GRF and RMSE values were higher in the X and Y directions but lower in the Z direction compared to the human-exoskeleton model, as shown in Table 3. The relatively low correlation of GRF Y may be due to its lower signal-to-noise ratio than the bionic model. The relatively low GRF X coefficients and GRF peaks can also be explained by the low signal-to-noise ratio due to the low amplitude. The short support phase time of the SSP subject-exoskeleton model is also an important factor.

**Table 3.** The average (standard deviation) of the Pearson correlation (*R*) and *RMSE* (N/KG) between the measured and predicted GRF and COP trajectory of the two models.

| Model | GRF X | | GRF Y | | GRF Z | | COP X | | COP Y | |
|---|---|---|---|---|---|---|---|---|---|---|
| | *R* | *RMSE* | *R* | *RMSE* | *R* | *RMSE* | *R* | *RMSE* | *R* | *RMSE* |
| Human-exoskeleton model | 0.795 (0.08) | 0.193 (0.06) | 0.857 (0.07) | 0.255 (0.06) | 0.974 (0.02) | 0.553 (0.23) | 0.325 (0.28) | 1.714 (0.36) | 0.743 (0.17) | 4.245 (3.04) |
| SSP subject-exoskeleton model | 0.478 (0.18) | 0.276 (0.06) | 0.536 (0.22) | 0.381 (0.08) | 0.943 (0.11) | 0.988 (0.47) | 0.135 (0.28) | 4.736 (3.86) | 0.398 (0.42) | 6.179 (2.63) |

The COP X trajectory of the exoskeletal model of the SSP subjects was right-sided. This is due to the limitation of the knee joint in the left lower limb. During the single support phase, the foot pressure was tilted to the right side to maintain equilibrium. The COP correlation was moderate. The root mean square error was relatively low. The error between measured and predicted COP trajectories may be due to low amplitude and therefore low signal-to-noise ratio, as well as the lack of edge node and exoskeletal segment deformation modeling.

## 6. Conclusions

In order to comprehensively analyze and evaluate the effects of the lower limb exoskeleton on the human body, two models were developed in AnyBody software, and a combination of simulation analysis and gait experiments was used. For the human exoskeleton model, the kinetic parameters satisfy the normal gait, but the muscle force increases abruptly. This is due to the sudden massive eccentric contraction of the muscles, and the size of the lower limb exoskeleton is not exactly the same as the human lower limb. For this reason, a stepless adjustable mechanism was designed. For the SSP subject-exoskeleton model, the RMSE of the GRF was low, with moderate to very high correlations, and the correlation of the COP trajectory was weak to moderate. Compared to the human exoskeleton model, the GRF was greater in all three directions, with the COP X trajectory more to the right and the COP Y trajectory more posterior.

In future studies, we will collaborate with rehabilitation hospitals. Using gait data from stroke patients, we will further explore the differences between normal people and stroke patients wearing exoskeleton robots. The model will enhance pressure detection at the edge of the foot. When comparing functional curves, we will use a procedure based on statistical parameter mapping to better describe the variability.

**Author Contributions:** The first author L.C. designed the experiment scheme, analyzed the experimental results, and wrote the paper. The author P.Z. and D.F. made gait experiments. The author J.Z. modified the paper. All authors have read and agreed to the published version of the manuscript.

**Funding:** This research was funded by Tianjin Science and Technology Program grant number 22YFZCSN00160.

**Institutional Review Board Statement:** Not applicable.

**Informed Consent Statement:** Informed consent was obtained from all subjects involved in the study.

**Data Availability Statement:** Not applicable.

**Conflicts of Interest:** The authors declare no conflict of interest.

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
