# Peer review of "Research on the Influence of Exoskeletons on Human Characteristics by Modeling and Simulation Using the AnyBody Modeling System"

_applsci, doi:10.3390/app13148184_

Round 1

Reviewer 1 Report

The article discusses the impact of exoskeleton usage on the kinetic and biomechanical parameters of both able-bodied individuals and patients with asymmetric motor disorders. The study develops two models and conducts experimental verification using virtual prototypes. Through the analysis of ground reaction force (GRF) and centre of pressure (COP) trajectories based on experimental data, the study demonstrates that wearing lower limb exoskeletons can enable walking gaits to meet the characteristics of normal human gait. However, there is a sudden increase in muscle strength in certain areas. The article provides support for optimizing the design of exoskeleton robots to make them more suitable for patients with asymmetric motor disorders. However, there are several issues with the article:

1. Abbreviation usage: The first occurrence of an abbreviation should be written in its full form, and subsequently, the abbreviation can be used.

2. During the experiments, the authors set the knee joint angle to 0 degrees for the five participants with motor disorders to clearly explore their walking characteristics. However, why wasn't the ankle joint dorsiflexion degree restricted?

3. Similarly, during the virtual prototype experiments, how was it ensured that the knee joint angle of the participants with motor disorders was 0 degrees? Were there any restrictions on the angles of the exoskeletons used?

4. The article introduces a novel mechanically adjustable exoskeleton mechanism but lacks detailed diagrams. It is recommended to include structural diagrams for better understanding.

5. The article has a limited number of references. It is suggested to include more relevant cutting-edge references.

6. The writing of certain parts of the article needs improvement.

In summary, the reviewer's suggestion is to consider accepting the article after the authors address and revise the mentioned points.

The writing of certain parts of the article needs improvement.

Author Response

Original Manuscript ID: applsci-2388935 

Original Article Title: “Research on Influence of Exoskeleton on Human Characteristics by Modelling and Simulating with AnyBody Modeling System

To: applied sciences-basel

Re: Response to reviewers

Dear Editor,

Thank you for allowing a re-submission of our manuscript, with an opportunity to address the reviewers’ comments.

We are uploading (a) our point-by-point response to the comments (below) (response to reviewers).

Best regards,

Junxia Zhang et al.

Reviewer#1, Concern # 1:

  1. Abbreviation usage: The first occurrence of an abbreviation should be written in its full form, and subsequently, the abbreviation can be used.

Author response: Agree

Author action: We corrected the problem.

Before modification: Methods: Virtual prototyping is a cost-effective method to validate the performance of exoskeleton robots. Therefore, two models, a human-exoskeleton model and an asymmetric movement disorder (SSP) subject-exoskeleton model, were developed in ANYBODY software for this study. The human-exoskeleton model was driven by kinematic data of 20 healthy participants walking in an exoskeleton at normal speed (3.6 km/h). As a comparison, the SSP subject-exoskeleton model was driven by data from 5 asymmetric movement disorder (SSP) subjects walking in an exoskeleton.

After modification: Methods: Virtual prototyping is a cost-effective method to validate the performance of exoskeleton robots. Therefore, two models, a human-exoskeleton model and an asymmetric movement disorder (SSP) subject-exoskeleton model, were developed in ANYBODY software for this study. The human-exoskeleton model was driven by kinematic data of 20 healthy participants walking in an exoskeleton at normal speed (3.6 km/h). As a comparison, the SSP subject-exoskeleton model was driven by data from 5 SSP subjects walking in an exoskeleton.

Reviewer#1, Concern # 2:

During the experiments, the authors set the knee joint angle to 0 degrees for the five participants with motor disorders to clearly explore their walking characteristics. However, why wasn't the ankle joint dorsiflexion degree restricted?

Author response: Agree

Author action: There is a problem with our description. In this paper, only the knee joint is constrained.

3.1 Participants

Twenty healthy volunteers (age: 24.70 ± 1.42 years old male, weight: 67.50 ± 8.59 kg, lower limb height: 900.30 ± 34.96 mm, knee joint width :112.49 ± 5.72 mm, and ankle joint width: 71.99 ± 4.15 mm) were recruited from Tianjin University of Science & Technology. Relatively young healthy subjects were selected for the gait analysis study, which can exclude the influence of age factor to the maximum extent and obtain more accurate and reliable study results, providing a better reference for the next study. In order to explore of the walking characteristics of SSP subjects clearly, the movement angle of knee joint was set as 0 degree for SSP subjects. The test was performed using an adjustable medical knee immobilization support (Medway, China, product size: 500 mm × 200 mm × 100 mm) and a thin wood strip with rounded corners (20 mm × 2.5 mm × 8 mm) to assist in tying the dorsal aspect of the knee joint, just as shown in Fig.3. The detailed component composition is shown in the Fig.4. The purpose was to limit the mobility of the knee joint in the flexion and extension directions to simulate the joint mobility characteristics of the knee joint in a rigid gait approximation. Written informed consent was obtained from the participants and this study was approved by the Tianjin University of Science and Technology Research Ethics Board. The study plan and experimental design meet ethical requirements, and the study is conducted with the informed consent of the participants.

(a) Front view                    (b) left view              (c) back view

Fig.3 Binding mode of knee fixation brace

Fig.4 Knee fixation brace, (A) Protective Velcro (B) Rounded corner support small wooden strips (C) lower leg velcro straps (D) thigh velcro straps (E) Left side aluminum support chuck (F) Right side aluminum support chuck

3.2 Gait experiments

The data collection platform is shown in Fig.5. The data collection platform includes Pedar-X (Novel Pedar, German), Telemyo 2400 DTS (Noraxon, America), AMTI three-dimensional force plates (AMTI-BP400600, America), and Vicon T40–S (Vicon, England). The frequency of data acquisition was all set at 1.5 kHz. The adjustable medical knee fixation support has an adjustment range of 0 to 180°. The specific steps of bondage are: (1) First, according to the condition of the patient's knee joint, select the appropriate product type to fit, and tie the magic pad; (2) The subject is then allowed to stand upright, with the axes of the angular chucks on each side corresponding to the lateral epicondyle of the knee on the wearing side; (3) Finally, fix the size of the leg straps and add rounded thin wooden strips to adjust the looseness. The subjects were allowed to walk for 10 min after wearing the device, during which time the subject's knee joint could not be flexed for adjustment and optimization. Due to the limitation of the knee joint of the left lower limb, the main objective of this study was to observe the kinematic and biomechanical changes in the right lower limb. Hence, the relevant parameters of the right lower limb in the sagittal plane were analyzed. The specific experiment scheme is as follows:

1). All participants were required to walk in a straight test bench (AMTI three-dimensional force plates), just as shown in Fig. 2(b).

2). To eliminate the error between different tests, the gait cycle was standardized. A complete gait cycle is 100%. The beginning of the gait cycle 0% means right heel first touches the ground. The end point 100% represents the right heel touches the ground again.

3). The walking test was repeated 5 times for each participant. Appropriate rest between sets was also given to exclude the effects of muscle fatigue.

Reviewer#1, Concern # 3:

Similarly, during the virtual prototype experiments, how was it ensured that the knee joint angle of the participants with motor disorders was 0 degrees? Were there any restrictions on the angles of the exoskeletons used?

Author response: Agree

Author action: We corrected some grammar errors and confusing sentences.

2.2 Musculoskeletal models construction

The lower limb exoskeleton model was imported into the AnyBody software, just as shown in the Fig. 2(a). It was controlled by program ‘FitnessMachine.any’. The human musculoskeletal model and the lower limb exoskeleton model were connected by program ‘JointsAndDrivers.any’. The human-exoskeleton model is shown in Fig. 2(b). The human musculoskeletal model was built based on the application repository model library. The motion data captured by Vicon system. The data was imported into AnyBody software through the "C3D to Anyscript conversion program". The inverse kinetic method was used to calculate muscle force, joint force and joint moments. The motion of the exoskeleton was constrained by the human kinematic data. The ankle and hip joints of the exoskeleton are aligned with the human body, respectively. The knee joint of the exoskeleton is aligned with the human body in the anterior-posterior axis. Compared to the human exoskeleton model, the freedom of flexion and extension of the left knee joint of SSP subject-exoskeleton model was removed.

Reviewer#1, Concern # 4:

The article introduces a novel mechanically adjustable exoskeleton mechanism but lacks detailed diagrams. It is recommended to include structural diagrams for better understanding.

Author response: Agree

Author action: We add the detailed diagrams.

2.1 The proposed novel lower limb exoskeleton robot

The biomechanics characteristics of human provides a reference for the structure design of the exoskeleton. The developed lower limb exoskeleton is shown in Fig. 1(a) and (b). The exoskeleton designed in this paper mainly considers the motion in sagittal plane. For each leg, it has one active DoF for hip joint and knee joint respectively, and one passive DoF for ankle joint. The limit of flexion in hip joint is 0°, the maximum extension angle is 70.18°. The limit of flexion in knee joint is 102.53°, the maximum extension angle is 0°. The hip joint and knee joint were driven by direct current servo motor, whereas the ankle joint adopted a passive reset function. The step function is utilized to control the speed of the ball screw. The linear motion of the ball screw is transformed into the rotational motion of the hip joint and knee joint in the sagittal plane, which approximately fits the normal gait of a human. This exoskeleton fits patients from 1.50m to 1.90m tall, which covers more than 99% of corresponding adults, with a maximum bodyweight of 100kg.

Fig. 1 Lower limb exoskeleton. (a) (1) Back support; (2) Driving source of hip joint; (3) Hip joint component; (4) Brace of thigh; (5) Driving source of knee joint;(6) Brace of calf; (7) Ankle joint component; (8) Flexible belt of waist;(9) Waist component;(10) Thigh component; (11) Flexible belt of thigh; (12) Knee joint component; (13) Calf component; (14) Flexible belt of calf; (15) Pedal; (b) Lower limb exoskeleton prototype

Reviewer#1, Concern # 5:

The article has a limited number of references. It is suggested to include more relevant cutting-edge references.

Author response: Agree

Author action: We add relevant cutting-edge references.

  1. Mohammad Sharif Shourijeh, M. Jung, Siu-Teing Ko, M. McGrath, N. Stech, M. Damsgaard, “Simulating Physiological Discomfort of Exoskeletons Using Musculoskeletal Modelling,” Gait & Posture. 2017. DOI: 10.1016/J.GAITPOST.2017.06.301
  2. Geonea I, Copilusi C, Margine A, et al. Dynamic Analysis and Structural Optimization of a New Exoskeleton Prototype for Lower Limb Rehabilitation[C]//International Workshop on Medical and Service Robots. Cham: Springer Nature Switzerland, 2023: 168-178.
  3. Vatsal V, Purushothaman B. Biomechanical Design Optimization of Passive Exoskeletons through Surrogate Modeling on Industrial Activity Data[C]//2022 IEEE/RSJ International Conference on Intelligent Robots and Systems (IROS). IEEE, 2022: 12752-12757.
  4. Dengiz F O. Research and Development on Mobile Powered Upper-Body Exoskeletons for Industrial Usage[C]//2022 21st International Symposium INFOTEH-JAHORINA (INFOTEH). IEEE, 2022: 1-6.
  5. Ji Zhongqiu. “Biomechanical characteristics of lower limbs in Taijiquan training simulated and verified by AnyBody,” Chinese Journal of rehabilitation medicine. vol. 66, no. 3, pp. 799-805, 2014.
  6. Shan Lijun, Hu Zhongan. “Analysis of muscle strength of hip joint rehabilitation training based on AnyBody,” Journal of Dalian Jiaotong University. vol. 78, no. 5, pp. 52-55, 2019.
  7. Xiang Zhongxia. “Training effect simulation of an exoskeleton rehabilitation robot,” Journal of Tianjin University. vol. 135, no. 23, pp. 698-701, 2016.
  8. Geonea I, Copilusi C, Margine A, et al. Dynamic Analysis and Structural Optimization of a New Exoskeleton Prototype for Lower Limb Rehabilitation[C]//International Workshop on Medical and Service Robots. Cham: Springer Nature Switzerland, 2023: 168-178.
  9. Wen Si. “The Crouch Start Modeling and Simulation Based on AnyBody Technology,” International Journal of Digital Content Technology and its Applications. vol. 4, no. 8, pp. 27-34, 2016. DOI: 10.4156 / jdcta.vol4.issue8.1

Reviewer#1, Concern # 6:

The writing of certain parts of the article needs improvement.

Author response: Agree

Author action: We corrected some grammar errors and confusing sentences.

Reviewer 2 Report

These findings will provide support for investigating the effects of the developed exoskeleton. Meanwhile, some characteristics of exoskeleton worn by asymmetric dyskinesia patients can also be preliminarily explored. Some comments given below.

1.      Explain in brief the challenges of rehabilitation with robot and manually with doctor.

2.      The rationalisation for not Reasonable design of exoskeleton robot should be provided.

3.      More comprehensive information support for figure 1 needs to provided.

4.      Several software used for similar purpose in the present study needs to resume in form of table for more better understanding.

5.      What is the current article novel? It has been extensively discussed in the past. Nothing truly novel in its current state. The absence of anything original makes the current study seem like a replication or a modified study. The introduction section should contain specifics about the writers' uniqueness. It is a significant reason to reject this study.

6.      The authors need to explain more clearly the basis of patient selection. Has any protocol, basis, or standard been followed? The present form was not unclear since the patient involved is heterogeneous with a small number. It would direct impact the results that lead to inappropriate work. One of the major issues that need to be solved by the authors after the revision stage.

7.      Step by step procedure provided in figure 2 needs more description.

8.      Related to modelling and simulating, the authors need to explain the urgency of this approach in the introduction section, such as bring the advantages in terms of lower cost and faster results compared to clinical and experimental study. Please provide this explanation along with relevant reference as follows: https://doi.org/10.3390/ma16093298, https://doi.org/10.3390/biomedicines11030951, and https://doi.org/10.1038/s41598-023-30725-6

9.      Related to gait data obtained, please give the information that this obtained data apart for exoskeleton development, is crucial for developing lower extremities medical implant. Please provide previous relevant reference that adopting gait as the loading condition as follows: https://doi.org/10.3390/ma14247554, https://doi.org/10.3390/su142013413, and https://jurnaltribologi.mytribos.org/v33/JT-33-31-38.pdf

-

Author Response

Original Manuscript ID: applsci-2388935 

Original Article Title: “Research on Influence of Exoskeleton on Human Characteristics by Modelling and Simulating with AnyBody Modeling System

To: applied sciences-basel

Re: Response to reviewers

Dear Editor,

Thank you for allowing a re-submission of our manuscript, with an opportunity to address the reviewers’ comments.

We are uploading our point-by-point response to the comments (below) (response to reviewers).

Best regards,

Junxia Zhang et al.

Reviewer#1, Concern # 1:

Explain in brief the challenges of rehabilitation with robot and manually with doctor.

Author response: Agree

Author action: We add the explanation.

According to information published in The Lancet, a leading medical journal, stroke has become the second leading cause of death worldwide [1-3]. The signs and symptoms after stroke onset are usually unilateral lower limb motor deficits [4]. The current approach to stroke rehabilitation relies on the joint participation of rehabilitation physicians and physical therapists, a rather lengthy and expensive process, and the most critical problem is the huge shortage of rehabilitation professionals [5]. However, rehabilitation robots can effectively and economically provide rehabilitation to physically disabled patients compared to the traditional manual assistance of rehabilitation physicians. The rehabilitation effects of lower limb rehabilitation robots have been widely recognized [6,7].

Reviewer#1, Concern # 2:

The rationalisation for not Reasonable design of exoskeleton robot should be provided.

Author response: Agree

Author action: First of all, the joint movement angle of the exoskeleton is in line with the movement pattern of the human body.

The biomechanics characteristics of human provides a reference for the structure design of the exoskeleton. The developed lower limb exoskeleton is shown in Fig. 1(a) and (b). The exoskeleton designed in this paper mainly considers the motion in sagittal plane. For each leg, it has one active DoF for hip joint and knee joint respectively, and one passive DoF for ankle joint. The limit of flexion in hip joint is 0°, the maximum extension angle is 70.18°. The limit of flexion in knee joint is 102.53°, the maximum extension angle is 0°. The hip joint and knee joint were driven by direct current servo motor, whereas the ankle joint adopted a passive reset function. The step function is utilized to control the speed of the ball screw. The linear motion of the ball screw is transformed into the rotational motion of the hip joint and knee joint in the sagittal plane, which approximately fits the normal gait of a human. This exoskeleton fits patients from 1.50m to 1.90m tall, which covers more than 99% of corresponding adults, with a maximum bodyweight of 100kg.

Fig. 1 Lower limb exoskeleton. (a) (1) Back support; (2) Driving source of hip joint; (3) Hip joint component; (4) Brace of thigh; (5) Driving source of knee joint;(6) Brace of calf; (7) Ankle joint component; (8) Flexible belt of waist;(9) Waist component;(10) Thigh component; (11) Flexible belt of thigh; (12) Knee joint component; (13) Calf component; (14) Flexible belt of calf; (15) Pedal; (b) Lower limb exoskeleton prototype

Secondly, Section 4.1 of the experimental results compares the different parameters with and without the exoskeleton. The experimental results show that the amplitude of the motion parameters fluctuate after wearing the exoskeleton, but the motion pattern is basically the same. This also indirectly justifies the exoskeleton robot.

Reviewer#1, Concern # 3:

More comprehensive information support for figure 1 needs to provided.

Author response: Agree

Author action: We add the detailed diagrams.

2.1 The proposed novel lower limb exoskeleton robot

The biomechanics characteristics of human provides a reference for the structure design of the exoskeleton. The developed lower limb exoskeleton is shown in Fig. 1(a) and (b). The exoskeleton designed in this paper mainly considers the motion in sagittal plane. For each leg, it has one active DoF for hip joint and knee joint respectively, and one passive DoF for ankle joint. The limit of flexion in hip joint is 0°, the maximum extension angle is 70.18°. The limit of flexion in knee joint is 102.53°, the maximum extension angle is 0°. The hip joint and knee joint were driven by direct current servo motor, whereas the ankle joint adopted a passive reset function. The step function is utilized to control the speed of the ball screw. The linear motion of the ball screw is transformed into the rotational motion of the hip joint and knee joint in the sagittal plane, which approximately fits the normal gait of a human. This exoskeleton fits patients from 1.50m to 1.90m tall, which covers more than 99% of corresponding adults, with a maximum bodyweight of 100kg.

Fig. 1 Lower limb exoskeleton. (a) (1) Back support; (2) Driving source of hip joint; (3) Hip joint component; (4) Brace of thigh; (5) Driving source of knee joint;(6) Brace of calf; (7) Ankle joint component; (8) Flexible belt of waist;(9) Waist component;(10) Thigh component; (11) Flexible belt of thigh; (12) Knee joint component; (13) Calf component; (14) Flexible belt of calf; (15) Pedal; (b) Lower limb exoskeleton prototype

Reviewer#1, Concern # 4:

Several software used for similar purpose in the present study needs to resume in form of table for more better understanding.

Author response: Agree

Author action: In this article, we mainly compare OpenSim and AnyBody in the introduction. Hence, we have made a detailed text description.

Musculoskeletal models are able to calculate muscle forces, joint reaction forces only from the input of a given motion. The field of musculoskeletal modeling has evolved a lot in the last two decades. There are many musculoskeletal modeling and simulation software available. Among the most popular ones are OpenSim and AnyBody Modeling System [11,12]. In the simulation system, the human body wears an exoskeleton to perform various movements and collects data on the interaction between the exoskeleton and the human to evaluate the performance of the exoskeleton in a more cost-effective and time-efficient manner. Ferrati used OpenSim to compare the changes in joint moments with and without the exoskeleton [13]. However, the simulations did not consider the properties of the muscles. Similarly, Zhu et al. used OpenSim and ADAMS to model the exoskeleton. The objective was to measure the alignment of the joints between the human body and the exoskeleton. A fitted kinetic equation was also developed to calculate the joint moments based on the virtual prototype [14]. Although OpenSim is open source, AnyBody is commercial software with an open code model repository, AnyBody Managed Model Repository, where users can use, modify, and contribute new models conveniently. Stambolian used AnyBody software to simulate the process of moving an object by the human body [15]. A comparative analysis of the reaction law of the object on the human body was performed by varying the weight of the object. Studies have used ADAMs and ANYBODY software to output joint torques of exoskeletons and to evaluate joint activity control performance [16-22]. Priyanshu et al. designed a virtual prototype set-up of a rehabilitation exoskeleton by merging computational musculoskeletal analysis and AnyBody simulation technology[16]. Agarwal based on AnyBody simulation to study and analyze the performance of the lower limb exoskeleton. The experimental results showed that enhancement through the exoskeleton can lead to a significant reduction in muscle loading[17]. In order to make the structure of the lower limb exoskeleton more compact and simple, and to provide a theoretical basis for the dynamic support effect of the lower limb exoskeleton, Li used AnyBody to simulate the knee joint lower limb exoskeleton. The simulation experiment results show that after wearing the lower limb exoskeleton, its energy consumption can be reduced by 58.6% when bearing an additional 100Kg payload. This proves that the knee-joint lower limb exoskeleton can effectively help the human body bear heavy loads and can reduce the energy consumption of the human body during the walking process [21].

Reviewer#1, Concern # 5:

What is the current article novel? It has been extensively discussed in the past. Nothing truly novel in its current state. The absence of anything original makes the current study seem like a replication or a modified study. The introduction section should contain specifics about the writers' uniqueness. It is a significant reason to reject this study.

Author response: Agree

Author action: We add the explanation.

Although many studies explored optimization strategies of exoskeleton through virtual simulation, there is a lack of human-machine simulation study of a real exoskeleton attached to the musculoskeletal model with unilateral movement disorders. At present, most studies have been performed around normal people. Most stroke patients exhibit unilateral dyskinesia [31-33]. Since exoskeletons are intended for patients with disabilities, it is necessary to explore whether exoskeletal robots could actually meet the needs of patients. In this paper, two virtual models were built in AnyBody software. The first one is a human-exoskeleton model driven by healthy slow walking kinematics. The kinetic and biomechanical parameters of the human model and the human exoskeleton model were compared and analyzed. The effect of the lower limb exoskeleton on the human body can be quantitatively analyzed. The second model is the SSP subject-exoskeleton model, which is driven by kinematic data of subjects with unilateral movement disorders walking with a real exoskeleton. Since the measurement data were obtained from participants wearing the exoskeleton, the predicted GRF accuracy will validate both models. The predicted GRF and COP trajectories of the human exoskeleton model and the SSP subject-exoskeleton model were also analyzed, which is important to investigate the differences between normal human and SSP subjects when wearing an exoskeleton at a normal speed of 3.6 km/h. The results of the study will support the optimal design of the exoskeleton robot to make it more suitable for SSP patients.

Reviewer#1, Concern # 6:

The authors need to explain more clearly the basis of patient selection. Has any protocol, basis, or standard been followed? The present form was not unclear since the patient involved is heterogeneous with a small number. It would direct impact the results that lead to inappropriate work. One of the major issues that need to be solved by the authors after the revision stage.

Author response: Agree

Author action: We add the explanation of subject selection.

Twenty healthy volunteers (age: 24.70 ± 1.42 years old male, weight: 67.50 ± 8.59 kg, lower limb height: 900.30 ± 34.96 mm, knee joint width :112.49 ± 5.72 mm, and ankle joint width: 71.99 ± 4.15 mm) were recruited from Tianjin University of Science & Technology. Relatively young healthy subjects were selected for the gait analysis study, which can exclude the influence of age factor to the maximum extent and obtain more accurate and reliable study results, providing a better reference for the next study.

Reviewer#1, Concern # 7:

Step by step procedure provided in figure 2 needs more description.

Author response: Agree

Author action: We add the description of figure 2.

2.2 Musculoskeletal models construction

The lower limb exoskeleton model was imported into the AnyBody software, just as shown in the Fig. 2(a). It was controlled by program ‘FitnessMachine.any’. The human musculoskeletal model and the lower limb exoskeleton model were connected by program ‘JointsAndDrivers.any’. The human-exoskeleton model is shown in Fig. 2(b). The human musculoskeletal model was built based on the application repository model library. The motion data captured by Vicon system. The data was imported into AnyBody software through the "C3D to Anyscript conversion program". The inverse kinetic method was used to calculate muscle force, joint force and joint moments. The motion of the exoskeleton was constrained by the human kinematic data. The ankle and hip joints of the exoskeleton are aligned with the human body, respectively. The knee joint of the exoskeleton is aligned with the human body in the anterior-posterior axis. Compared to the human exoskeleton model, the freedom of flexion and extension of the left knee joint of SSP subject-exoskeleton model was removed.

(b)                     (c)

Fig. 2 (a) Lower limb exoskeleton model; (b) Human-exoskeleton model

Reviewer#1, Concern # 8:

Please provide relevant references.

Author response: Agree

Author action: We add the references.

  1. Salaha, Z.F.M.; Ammarullah, M.I.; Abdullah, N.N.A.A.; Aziz, A.U.A.; Gan, H.-S.; Abdullah, A.H.; Abdul Kadir, M.R.; Ramlee, M.H. Biomechanical Effects of the Porous Structure of Gyroid and Voronoi Hip Implants: A Finite Element Analysis Using an Experimentally Validated Model. Materials 2023, 16, 3298. https://doi.org/10.3390/ma16093298
  2. Ammarullah, M.I.; Hartono, R.; Supriyono, T.; Santoso, G.; Sugiharto, S.; Permana, M.S. Polycrystalline Diamond as a Potential Material for the Hard-on-Hard Bearing of Total Hip Prosthesis: Von Mises Stress Analysis. Biomedicines 2023, 11, 951. https://doi.org/10.3390/biomedicines11030951
  3. Tauviqirrahman, M., Ammarullah, M.I., Jamari, J. et al. Analysis of contact pressure in a 3D model of dual-mobility hip joint prosthesis under a gait cycle. Sci Rep 13, 3564 (2023). https://doi.org/10.1038/s41598-023-30725-6
  4. Ammarullah, M.I.; Afif, I.Y.; Maula, M.I.; Winarni, T.I.; Tauviqirrahman, M.; Akbar, I.; Basri, H.; van der Heide, E.; Jamari, J. Tresca Stress Simulation of Metal-on-Metal Total Hip Arthroplasty during Normal Walking Activity. Materials 2021, 14, 7554. https://doi.org/10.3390/ma14247554
  5. Ammarullah, M.I.; Santoso, G.; Sugiharto, S.; Supriyono, T.; Wibowo, D.B.; Kurdi, O.; Tauviqirrahman, M.; Jamari, J. Minimizing Risk of Failure from Ceramic-on-Ceramic Total Hip Prosthesis by Selecting Ceramic Materials Based on Tresca Stress. Sustainability 2022, 14, 13413. https://doi.org/10.3390/su142013413
  6. Muhammad Imam Ammarullah 1,2* , Gatot Santoso 1, S. Sugiharto 1, Toto Supriyono 1, Ojo Kurdi 3, Mohammad Tauviqirrahman 3, Tri Indah Winarni 2,4,5, J. Jamari. Tresca stress study of CoCrMo-on-CoCrMo bearings based on body mass index using 2D computational model. Jurnal Tribologi, 2022, 33, 31-38. https://jurnaltribologi.mytribos.org/v33/JT-33-31-38.pdf.

Round 2

Reviewer 2 Report

Well effort in the authors so far. Some correction still needed as follows:

1.      In page 1, I am concern regarding the initial information provided “According to information published in The Lancet, a leading medical journal”. It would be better to change this statement for avoiding misunderstanding.

2.      In page 2, the authors explain that ADAMs and Anybody software have been widely used in simulated joint condition. Please give the rationalisation. Is the nothing any other competitive software apart from both? Or both of the have some advantages comparing to others.

3.      In page 4, please make Figure 1 into component a and b for more better presentation.

4.      In page 4 for caption of Figure 1, please explain the component using table rather than become bulky in the caption.

5.      In page 4, please recheck the caption in Figure 2 and its figure, The reviewer think the auhtors made an error with using (c) that instead of (b).

6.      Loading from lower limb has been incorporated for several computational simulation approach as done by Jamari et al. Please explain this statement and support by relevant reference as follows: https://doi.org/10.3390/met12081241, https://doi.org/10.3390/jfb13020064, and https://doi.org/10.1016/j.heliyon.2022.e12050

7.      Modelling and simulation have been widely used in the field of engineering, medicine, and physics to evaluate complex parameter starting from load, condition, material, and others parameter. Please provide this information for reader better understanding for urgency of modelling and simulation. Also, refer the relevant reference as follows: https://doi.org/10.3390/jfb12020038, https://doi.org/10.3390/pr11051540, https://doi.org/10.1177/14657503221144810, https://doi.org/10.3390/fluids7070225, and https://doi.org/10.1080/23311916.2023.2218691

-

Author Response

Original Manuscript ID: applsci-2388935 

Original Article Title: “Research on Influence of Exoskeleton on Human Characteristics by Modelling and Simulating with AnyBody Modeling System

To: applied sciences-basel

Re: Response to reviewers

Dear Editor,

Thank you for allowing a re-submission of our manuscript, with an opportunity to address the reviewers’ comments.

We are uploading our point-by-point response to the comments (below) (response to reviewers).

Best regards,

Junxia Zhang et al.

Reviewer#1, Concern # 1:

In page 1, I am concern regarding the initial information provided “According to information published in The Lancet, a leading medical journal”. It would be better to change this statement for avoiding misunderstanding.

Author response: Agree

Author action: We revise the explanation.

Stroke is characterized by a high incidence and disability rate [1-3], which can place a heavy burden on patients and their families.

Reviewer#1, Concern # 2:

In page 2, the authors explain that ADAMs and Anybody software have been widely used in simulated joint condition. Please give the rationalisation. Is the nothing any other competitive software apart from both? Or both of the have some advantages comparing to others.

Author response: Agree

Author action: We used OpenSim software as a comparison and briefly introduced the advantages of AnyBody software.

Compared to OpenSim software, AnyBody has an open code model library. With the AnyBody model library, users can use and modify new models more easily. This greatly simplifies modeling.

Reviewer#1, Concern # 3:

In page 4, please make Figure 1 into component a and b for more better presentation.

Author response: Agree

Author action: We add the detailed introduction of the exoskeleton.

2.1 The proposed novel lower limb exoskeleton robot

The biomechanics characteristics of human provides a reference for the structure design of the exoskeleton. The developed lower limb exoskeleton is shown in Fig. 1(a) and (b). The introduction of each component of the lower limb exoskeleton robot is shown in table 1. The exoskeleton designed in this paper mainly considers the motion in sagittal plane. For each leg, it has one active DoF for hip joint and knee joint respectively, and one passive DoF for ankle joint. The limit of flexion in hip joint is 0°, the maximum extension angle is 70.18°. The limit of flexion in knee joint is 102.53°, the maximum extension angle is 0°. The hip joint and knee joint were driven by direct current servo motor, and the parameter values of joint power elements are shown in table 2. The ankle joint adopted a passive reset function. The step function is utilized to control the speed of the ball screw. The linear motion of the ball screw is transformed into the rotational motion of the hip joint and knee joint in the sagittal plane, which approximately fits the normal gait of a human. This exoskeleton fits patients from 1.50m to 1.90m tall, which covers more than 99% of corresponding adults, with a maximum bodyweight of 100kg.

(a)                       (b)

Fig. 1 Lower limb exoskeleton. (a) (1) Back support; (2) Driving source of hip joint; (3) Hip joint component; (4) Brace of thigh; (5) Driving source of knee joint;(6) Brace of calf; (7) Ankle joint component; (8) Flexible belt of waist;(9) Waist component;(10) Thigh component; (11) Flexible belt of thigh; (12) Knee joint component; (13) Calf component; (14) Flexible belt of calf; (15) Pedal; (b) Lower limb exoskeleton prototype

Table 1. Introduction of each component of the lower limb exoskeleton robot

Component

Material

Mass/Kg

Back support

Carbon fiber

1.2

Hip joint component

High strength aluminum alloy

1.5

Knee joint component

High strength aluminum alloy

1.2

Ankle joint component

High strength aluminum alloy

1.2

Brace of thigh

Carbon fiber

2.4

Brace of calf

Carbon fiber

2.4

Table 2. Parameter values of joint power elements

Power Components

Specification

Hip joint

Knee joint

Motor

Rated output (W)

750

400

Rated torque(N-m)

2.4

1.3

Ball Screws

Stroke range (mm)

100

100

Maximum dynamic load (N)

4000

2000

Reviewer#1, Concern # 4:

In page 4, please recheck the caption in Figure 2 and its figure, The reviewer think the auhtors made an error with using (c) that instead of (b).

Author response: Agree

Author action: We corrected the problem.

Reviewer#1, Concern # 5:

Loading from lower limb has been incorporated for several computational simulation approach as done by Jamari et al. Please explain this statement and support by relevant reference as follows: https://doi.org/10.3390/met12081241, https://doi.org/10.3390/jfb13020064, and https://doi.org/10.1016/j.heliyon.2022.e12050.

Modelling and simulation have been widely used in the field of engineering, medicine, and physics to evaluate complex parameter starting from load, condition, material, and others parameter. Please provide this information for reader better understanding for urgency of modelling and simulation. Also, refer the relevant reference as follows: https://doi.org/10.3390/jfb12020038, https://doi.org/10.3390/pr11051540, https://doi.org/10.1177/14657503221144810, https://doi.org/10.3390/fluids7070225, and https://doi.org/10.1080/23311916.2023.2218691

Author response: Agree

Author action: We add the references.

  1. Jamari, J.; Ammarullah, M.I.; Santoso, G.; Sugiharto, S.; Supriyono, T.; van der Heide, E. In Silico Contact Pressure of Metal-on-Metal Total Hip Implant with Different Materials Subjected to Gait Loading. Metals 2022, 12, 1241. https://doi.org/10.3390/met12081241
  2. Jamari, J.; Ammarullah, M.I.; Santoso, G.; Sugiharto, S.; Supriyono, T.; Prakoso, A.T.; Basri, H.; van der Heide, E. Computational Contact Pressure Prediction of CoCrMo, SS 316L and Ti6Al4V Femoral Head against UHMWPE Acetabular Cup under Gait Cycle. J. Funct. Biomater. 2022, 13, 64. https://doi.org/10.3390/jfb13020064
  3. J. Jamari, Muhammad Imam Ammarullah, Gatot Santoso, S. Sugiharto, Toto Supriyono, Muki Satya Permana, Tri Indah Winarni, Emile van der Heide, Adopted walking condition for computational simulation approach on bearing of hip joint prosthesis: review over the past 30 years, 2022, 8.
  4. Jamari, J.; Ammarullah, M.I.; Saad, A.P.M.; Syahrom, A.; Uddin, M.; van der Heide, E.; Basri, H. The Effect of Bottom Profile Dimples on the Femoral Head on Wear in Metal-on-Metal Total Hip Arthroplasty. J. Funct. Biomater. 2021, 12, 38. https://doi.org/10.3390/jfb12020038
  5. Mughal, K.; Mughal, M.P.; Farooq, M.U.; Anwar, S.; Ammarullah, M.I. Using Nano-Fluids Minimum Quantity Lubrication (NF-MQL) to Improve Tool Wear Characteristics for Efficient Machining of CFRP/Ti6Al4V Aeronautical Structural Composite. Processes 2023, 11, 1540. https://doi.org/10.3390/pr11051540
  6. Tauviqirrahman, M.; Jamari, J.; Susilowati, S.; Pujiastuti, C.; Setiyana, B.; Pasaribu, A.H.; Ammarullah, M.I. Performance Comparison of Newtonian and Non-Newtonian Fluid on a Heterogeneous Slip/No-Slip Journal Bearing System Based on CFD-FSI Method. Fluids 2022, 7, 225. https://doi.org/10.3390/fluids7070225
  7. Lamura MDP, Hidayat T, Ammarullah MI, Bayuseno AP, Jamari J. Study of contact mechanics between two brass solids in various diameter ratios and friction coefficient. Proceedings of the Institution of Mechanical Engineers, Part J: Journal of Engineering Tribology. 2023;0(0). doi:10.1177/14657503221144810
  8. M. Danny Pratama Lamura, Muhammad Imam Ammarullah, Taufiq Hidayat, Mohamad Izzur Maula, J. Jamari & Athanasius Priharyoto Bayuseno (2023) Diameter ratio and friction coefficient effect on equivalent plastic strain (PEEQ) during contact between two brass solids, Cogent Engineering, 10:1, DOI: 10.1080/23311916.2023.2218691
